# Prevalence of high screen time and associated factors among students: a cross-sectional study in Zhejiang, China

Hao Wang,[1] Jieming Zhong,[1] Ruying Hu,[1] Bragg Fiona,[2] Min Yu,[1] Huaidong Du[2,3]

[1]Department of NCDs Control and Prevention, Zhejiang Provincial Center for Diseases Control and Prevention, Hangzhou, China
[2]Clinical Trial Service Unit and Epidemiological Studies Unit (CTSU), Nuffield Department of Population Health, University of Oxford, Oxford, UK
[3]Medical Research Council Population Health Research Unit, Nuffield Department of Population Health, University of Oxford, Oxford, UK

**Correspondence to**
Professor Min Yu;
myu@cdc.zj.cn

## ABSTRACT

**Objective** To investigate the prevalence and correlates of high screen time (ST) among students in Zhejiang, China.
**Design** Cross-sectional study.
**Setting** School-based adolescent health survey in Zhejiang Province, China.
**Participants** 23 543 students in grades 7–12 from 442 different schools.
**Outcome** High ST.
**Results** The mean age of the students was 15.6 years and 49.7% of them were girls. The prevalence of high ST (screen viewing ≥2 hours per day) was 42.4% (95% CI 40.2% to 44.5%), higher in boys than in girls (45.4% (95% CI 42.8% to 48.0%) vs 39.1% (95% CI 36.6% to 41.7%)). No statistically significant difference was found between urban and rural areas (43.0% (95% CI 37.2% to 48.7%) vs 42.1% (95% CI 39.6% to 44.6%)). The prevalence of high ST among middle school, academic high school and vocational high school students was 35.3%, 30.0% and 73.5%, respectively. Multivariable logistic analysis showed that older age, attendance at vocational high school, non-intact family, poor academic performance, bad self-reported health status, loneliness and drinking carbonated beverages ≥3 times every day were positively associated with high ST. Attendance at academic high school, higher parental education and being physically active were negatively associated with high ST.
**Conclusions** High ST was prevalent among students and associated with a cluster of sociodemographic and behavioural risk factors in Zhejiang, China.

## Strengths and limitations of this study

- This is a school-based study with a representative sample from provincial China, a high response rate and a standardised procedure.
- The study questionnaire covers a range of socio-demographic and behavioural risk factors, and the findings provide evidence to support health, and other, professionals in formulating intervention strategies to control screen time (ST).
- The cross-sectional study design prevents establishment of causal relationships between sociodemographic and behavioural factors and high ST.
- Detailed information about ST (such as duration of watching television, computer use and playing electronic games) is not available in this study.

## INTRODUCTION

Rapid economic development over recent decades has been accompanied by dramatic transitions in lifestyles in China. Prolonged sedentary time and low physical activity are becoming more common in China,[1] and exposure to electronic screen products is known to be the most population sedentary leisure activity among adolescents. At the same time, prevalences of obesity and diabetes have increased dramatically in the Chinese population.[2–4]

According to a report released by China Internet Information Center,[5] the number of internet users in China increased from 0.54 billion to 0.71 billion during 2012–2017,

and the proportion of the population using mobile internet devices increased from 72.2% to 92.5% over the same period. Compared with traditional desktop computers, smartphones provide an easier means for adolescents to access the internet, resulting in more time spent on electronic screen products. In the USA, while the proportion of high school students exceeding the recommended 2 or fewer hours/day of television (TV) viewing time decreased significantly from 43% to 32% between 1999 and 2013, the proportion who spent more than 2 hours per day playing video or computer games nearly doubled from 22% to 41%.[6] This transition may also happen in China in the near future, where adolescents aged 10–19 years account for approximately 20% of the total population of China.[5]

In China, approximately 60% of inner-city adolescents are estimated to have one or more screen products in their bedrooms.[7] High screen time (ST) among adolescents has been increasingly recognised as a serious public health concern and this continues into early adulthood.[8] Previous studies have indicated that high ST is associated with chronic diseases (eg, obesity, metabolic syndrome),[9–12] and with adolescents' psychological health.[13] One study found that watching TV or using a computer

for more than 3 hours a day was positively associated with health complaints (eg, headache, low mood, irritability and anxiety), and these associations were not mediated by low physical activity levels.[14] Another European study found a positive relationship between high ST and school problems (eg, truancy and poor academic performance).[15] Evidence to date on sociodemographic and lifestyle correlates of high ST comes mainly from Western countries,[16–18] with little known about these associations in China. Although a previous Chinese study, including 5003 adolescents found a high ST prevalence of 26%,[19] the study included only middle school and not high school, students. The study was conducted in 2010, and, during the past 7 years, technology has developed rapidly and screen products have become more widespread globally. It is important, therefore, to examine the prevalence of high ST and its correlating factors among adolescents in China.

## METHODS

### Sample and procedure

A cross-sectional study was carried out between April and May 2017 in Zhejiang province, China, using a three-stage sampling design. In stage 1, 30 counties, including 12 urban areas and 18 rural areas, were sampled randomly from all 90 counties in Zhejiang. In stage 2, 10 classes of middle school, 5 classes of academic high school and 5 classes of vocational high school were selected randomly within each chosen county. In stage 3, all students attending the chosen classes were invited to participate in the survey. Written informed consent was obtained from all participants and their guardians before the survey. A total of 24 157 students from grades 7 to 12 in 442 different schools were invited to participate. A response rate of 97.5% was achieved, and after exclusion of individuals with missing or incomplete questionnaires, 23 543 participants were included in the final analyses. A total of 12 068 (51.3%) were boys and the overall mean age was 15.6 years. A total of 12 207 (51.9%) students were from middle schools, 6477 (27.5%) from academic high schools and 4859 (20.6%) from vocational high schools. The survey questionnaire was based on the Youth Risk Behaviour Survey, developed by the Centers for Disease Control and Prevention (CDC)[20] and the international Global School-based Student Health Survey supported by WHO.[21] The reliability of questionnaire has been reported in previous studies.[22–24] The questionnaire covered demographic characteristics, tobacco and alcohol use, physical activity, dietary habit, exposure to violence, injury and sexual behaviours. The self-administrated questionnaire was filled anonymously by students and put directly into sealed boxes after completion. In order to improve response rate, every recruited student was given a pencil box as a gift.

### Patient and public involvement

Study participants were generally healthy students and no patients were involved in the study. Students and their parents were not involved in the design and conduct of study. The findings will be disseminated to Department of Health and Department of Education in Zhejiang Province, but not directly to participating students.

### Measures

#### Outcome variable

ST was assessed through the question: 'On an average school day, how many hours do you watch TV, play pad or electronic games or use a smartphone or computer for something that is not school work?' (Answer options: 'I do not watch TV, play pad or electronic games or use a smartphone or computer for something that is not school work', '<1 hour/day', '1 hour/day', '2 hours/day', '3 hours/day', '4 hours/day' and '≥5 hours/day'). Participants were considered as high ST users if they answered that they had watched a screen for more than 2 hours on an average school day.[25 26]

#### Main covariates

Information was collected on parental education level, parental marital status, academic performance, loneliness, physical activity, breakfast behaviour and intake of fruit, vegetables and carbonated beverages (table 1).

### Statistical analysis

A weighting factor was applied to each student record to adjust for non-response and for the varying probabilities of selection. The weight used for estimation in this survey was given by: W=W1×W2×f1×f2, where W1=the inverse of the probability of selecting the county; W2=the inverse of the probability of selecting the classroom within the county; f1=a student-level non-response adjustment factor calculated by class; f2=a poststratification adjustment factor calculated by grade.[27] Continuous variables were shown as mean±SD deviation. Prevalence of high ST was estimated as percentage with its 95% CI. Between group comparisons of categorical variables were undertaken using the $X^2$ test. Weighted prevalence between groups was calculated using the Rao-Scott $X^2$ test. Multivariable logistic regression was used to ascertain factors related to high ST. All analyses were performed with SAS software V.9.3. All statistical tests were two tailed, with p values<0.05 considered statistically significant.

## RESULTS

### Descriptive statistics

37.7% boys and 39.0% girls were from urban areas (table 2). As compared with girls, boys were more likely to describe their personal health status as very good or good (56.0% vs 49.3%) and less likely to feel lonely (33.0% vs 38.7%). 16.7% of students reported being physically active every day. 70.6% of students reported consuming breakfast every day. 28.7% and 8.0% of students reported consuming fruit and vegetables, respectively, less than once daily. There was no sex difference in the frequency of carbonated beverages consumption (p=0.19).

**Table 1** Questions and answer options included in the survey

| Variables | Questions | Answer options |
|---|---|---|
| Parental education level | What is the highest level of education your father/mother has obtained? (separately for father and mother | Primary school or below, middle school, high school, college or university, master graduates or above, unknown. |
| Parental marital status | What is your parents current marital status? | Married, divorced, widowed, separated. |
| Academic performance | How would you describe your grades in your class? | Excellent, middle, poor. |
| Self-reported health | In general, how would you describe your health status? | Very good, good, fair, bad, very bad and unknown |
| Loneliness | During the past 12 months, did you ever feel lonely? | Never, occasionally, sometimes, often, always. |
| Physical activity | During the past 7 days, on how many days were you physically active for a total of at least 60 min per day? | None, 1 day, 2 days, 3 days, 4 days, 5 days, 6 days, 7 days. |
| Breakfast | During the past 7 days, on how many days did you eat breakfast? | 0 days, 1 day, 2 days, 3 days, 4 days, 5 days, 6 days, 7 days. |
| Fruit | During the past 30 days, how many times per day did you usually eat fruit, such as apples, oranges, mangoes or papayas? | None, <1 time/day, 1 time/day, 2 times/day, 3 times/day, 4 times/day, ≥5 times/day. |
| Vegetable | During the past 30 days, how many times per day did you usually eat vegetables, such as cauliflower, cabbage? | None, <1 time/day, 1 time/day, 2 times/day, 3 times/day, 4 times/day, ≥5 times/day. |
| Carbonated beverages | During the past 30 days, how many times per day did you usually drink carbonated soft drinks, such as Coca-Cola, Pepsi or Sprite? (Do not include diet soft drinks.) | None, 1-3 times/week, 4-6 times/week, 1 time/day, 2 times/day, 3 times/day, ≥4 times/day. |

### The prevalence of high ST

The overall prevalence of high ST was 42.4% (95% CI 40.2% to 44.5%), higher in boys than in girls (45.4% vs 39.1%) (table 3). There was no statistically significant difference in high ST prevalence between urban and rural areas (43.0% vs 42.1%). Prevalence of high ST was positively associated with age (p<0.0001). The prevalence of high ST among students attending middle, academic high and vocational high school was 35.3%, 30.0% and 73.5%, respectively.

### Logistic regression analysis

After adjusting for all other sociodemographic and health-related behavioural factors under investigation, parental education level was inversely associated with high ST (table 4). Students whose fathers were educated to college level or above had 31% lower (OR 0.69, 95% CI 0.58 to 0.82) risk of high ST compared with students whose fathers were educated to middle school level or below. Similar associations were seen for maternal education level; students whose mothers were educated to college level or above had 34% lower risk of high ST (OR 0.66, 95% CI 0.58 to 0.76) compared with students whose mothers were educated to middle school level or below. Students living in non-intact families had a 26% higher risk of high ST (OR 1.26, 95% CI 1.13 to 1.41) than students living in intact families, and those with bad academic performance were 2.1 times more likely to report high ST than those with excellent academic performance (OR 2.07, 95% CI 1.86 to 2.30). Compared with students with very good or good self-reported health, students with fair, bad or very bad self-reported health had 14% (OR 1.14, 95% CI 1.07% to 1.22%) and 31% (OR 1.31, 95% CI 1.14% to 1.49%) higher risk of high ST, respectively. Students who often or always felt lonely were 20% (OR 1.20, 95% CI 1.08% to 1.34%) more likely to report high ST than those who never or occasionally felt lonely. Compared with students who were not physically active within the past 7 days, those who were physically active had 10% lower risk of high ST (OR 0.90, 95% CI 0.81 to 0.99). Fruit consumption was not associated with ST in a linear association; compared with students who reported eating fruit 1–2 times per day, those who reported eating fruit less than one daily and ≥3 times per day had 44% (OR 1.44, 95% CI 1.29% to 1.60%) and 18% (OR 1.18, 95% CI 1.01% to 1.38%) higher risks of high ST, respectively. Compared with students who did not consume carbonated beverages, those who reported consuming carbonated beverages ≥3 times per day had 29% higher odds of high ST (OR 1.29, 95% CI 1.03 to 1.60).

### DISCUSSION

Through a provincial representative survey among students in Zhejiang, China, our study investigated the

| Characteristics | Total (n=23 543) | Boys (n=12 068, 51.3%) | Girls (n=11 475, 49.7%) | P values |
|---|---|---|---|---|
| **Age (years)** | | | | 0.577 |
| ≤13 | 5159 (21.9) | 2689 (22.3) | 2470 (21.5) | |
| 14 | 4300 (18.3) | 2192 (18.1) | 2108 (18.4) | |
| 15 | 3730 (15.8) | 1905 (15.8) | 1825 (15.9) | |
| ≥16 | 10 354 (44.0) | 5282 (43.8) | 5072 (44.2) | |
| **Area** | | | | 0.031 |
| Urban | 9022 (38.3) | 4544 (37.7) | 4478 (39.0) | |
| Rural | 14 521 (61.7) | 7524 (62.3) | 6997 (61.0) | |
| **Types of school** | | | | 0.008 |
| Middle school | 12 207 (51.8) | 6364 (52.7) | 5843 (50.9) | |
| Academic high school | 6477 (27.5) | 3223 (26.7) | 3254 (28.4) | |
| Vocational high school | 4859 (20.6) | 2481 (20.6) | 2378 (20.7) | |
| **Paternal education** | | | | 0.0136 |
| Middle or below | 13 568 (57.6) | 6908 (57.2) | 6660 (58.0) | |
| High school | 5100 (21.7) | 2628 (21.8) | 2472 (21.5) | |
| College or above | 3129 (13.3) | 1575 (13.1) | 1554 (13.5) | |
| Unknown | 1746 (7.4) | 957 (7.9) | 789 (7.0) | |
| **Maternal education** | | | | <0.0001 |
| Middle or below | 14 530 (61.7) | 7292 (60.4) | 7238 (63.1) | |
| High school | 4363 (18.5) | 2271 (18.8) | 2092 (18.2) | |
| College or above | 2736 (11.6) | 1392 (11.5) | 1344 (11.7) | |
| Unknown | 1914 (8.1) | 1113 (9.2) | 801 (7.0) | |
| **Parental marital status** | | | | 0.0004 |
| Married | 21 151 (89.8) | 10 924 (90.5) | 10 227 (89.1) | |
| Other | 2392 (10.2) | 1144 (9.5) | 1 248 (10.9) | |
| **Academic performance** | | | | <0.0001 |
| Excellent | 5448 (23.1) | 2731 (22.6) | 2717 (23.7) | |
| Middle | 11 765 (50.0) | 5727 (47.5) | 6038 (52.6) | |
| Poor | 6330 (26.9) | 3610 (29.9) | 2720 (23.7) | |
| **Self-reported health** | | | | <0.0001 |
| Very good/good | 12 415 (52.7) | 6758 (56.0) | 5657 (49.3) | |
| Fair | 9563 (40.6) | 4495 (37.2) | 5068 (44.2) | |
| Very bad/bad | 1293 (5.5) | 650 (5.4) | 643 (5.6) | |
| Unknown | 272 (1.2) | 165 (1.4) | 107 (0.9) | |
| **Loneliness** | | | | <0.0001 |
| Never/occasionally | 15 122 (64.2) | 8082 (67.0) | 7040 (61.3) | |
| Sometimes | 5783 (24.6) | 2698 (22.4) | 3085 (26.9) | |
| Often/always | 2638 (11.2) | 1288 (10.6) | 1350 (11.8) | |
| **Physical activity (day/week)** | | | | <0.0001 |
| 0 | 4883 (20.7) | 2079 (17.2) | 2804 (24.5) | |
| 1–2 | 5690 (24.2) | 2703 (22.4) | 2987 (26.0) | |
| 3–5 | 8050 (34.2) | 4237 (35.1) | 3813 (33.2) | |
| 6–7 | 4920 (20.9) | 3049 (25.3) | 1871 (16.3) | |
| **Breakfast (day/week)** | | | | <0.0001 |

Table 2  Characteristics of students from Zhejiang

**Table 2** Continued

| Characteristics | Total (n=23 543) | Boys (n=12 068, 51.3%) | Girls (n=11 475, 49.7%) | P values |
|---|---|---|---|---|
| 0 | 473 (2.0) | 289 (2.39) | 184 (1.60) | |
| 1–2 | 599 (2.5) | 298 (2.47) | 301 (2.62) | |
| 3–4 | 1249 (5.3) | 667 (5.53) | 582 (5.07) | |
| ≥5 | 21 222 (90.2) | 10 814 (89.6) | 10 408 (90.7) | |
| Fruit (times/day) | | | | 0.01 |
| ≥3 | 6847 (29.1) | 3453 (28.6) | 3394 (29.6) | |
| 1–2 | 9945 (42.2) | 5213 (43.2) | 4732 (41.2) | |
| <1 | 6751 (28.7) | 3402 (28.2) | 3349 (29.2) | |
| Vegetable (times/day) | | | | 0.40 |
| ≥3 | 11 775 (50.0) | 5984 (49.6) | 5791 (50.5) | |
| 1–2 | 9884 (42.0) | 5108 (42.3) | 4776 (41.6) | |
| <1 | 1884 (8.0) | 976 (8.1) | 908 (7.9) | |
| Carbonated beverages | | | | 0.19 |
| None | 9133 (38.8) | 4717 (39.1) | 4416 (38.5) | |
| 1–6 times/week | 12 792 (54.3) | 6529 (54.1) | 6263 (54.6) | |
| 1–2 times/day | 1146 (4.9) | 600 (5.0) | 546 (4.7) | |
| ≥3 times/day | 472 (2.0) | 222 (1.8) | 250 (2.2) | |

prevalence of high ST associated with several sociodemographic (eg, parental education level and marital status) and behavioural factors (eg, inadequate physical activity, skipping breakfast, insufficient intake of fruits, drinking carbonated beverages). The findings provide evidence to

**Table 3** Weighted prevalence of high screen time among students from Zhejiang by different characteristics

| Characteristics | Prevalence (%)* | Rao-Scott $\chi^2$ | P values |
|---|---|---|---|
| Age (year) | | 89.05 | <0.0001 |
| ≤13 | 31.0 (27.4–34.7) | | |
| 14 | 36.4 (33.6–39.3) | | |
| 15 | 43.5 (40.2–46.9) | | |
| ≥16 | 49.6 (46.0–53.1) | | |
| Sex | | 18.03 | <0.0001 |
| Boys | 45.4 (42.8–48.0) | | |
| Girls | 39.1 (36.6–41.7) | | |
| Areas | | 0.06 | 0.81 |
| Urban | 43.0 (37.2–48.7) | | |
| Rural | 42.1 (39.6–44.6) | | |
| Types of school | | 404.57 | <0.0001 |
| Middle school | 35.3 (33.0–37.6) | | |
| Academic high school | 30.0 (27.2–32.9) | | |
| Vocational high school | 73.5 (69.5–77.5) | | |

*Based on the weighted data.

support the development and implementation of policies or interventions to control ST among middle and high school students in Zhejiang.

### The prevalence of high ST

The use of different questionnaires to evaluate ST in this and previous studies makes direct comparison of the results difficult. In our study more than 40% students exceeded the recommended maximum ST of 2 hours/day, suggesting that excessive exposure to electronic screen products is becoming more common among adolescents in China. A study using a similar questionnaire as that used in the current study and conducted in Brazil in 2013–2014 showed that 59.5% of students aged 12–17 years were exposed to electronic screens for ≥2 hours/day,[28] higher than our study. Consistent with results from other studies, we found that boys had higher prevalence of high ST than girls[7 19 29] and the prevalence of high ST increased with increasing age.[29 30] The gender difference might be explained by the fact that boys tend to be more attracted to computer games (such as sports, racing, fighting, shooting) than girls.[31] Another possible reason might be that girls usually spend more time on homework than boys in China. Notably, among students attending three different types of school, those attending vocational high schools had the highest prevalence of high ST, suggesting students of vocational high school may be an appropriate target population for interventions to reduce the prevalence of high ST. A possible explanation for differences between school types might be that vocational high school students do not face competitive high school or college entrance examinations, unlike students

**Table 4** CORs and AORs for high screen time in relation to sociodemographic and behavioural factors among students from Zhejiang

| Characteristics | COR (95% CI) | AOR (95% CI)† |
|---|---|---|
| Age (ref: ≤13 years) | | |
| 14 years | 1.27 (1.06 to 1.54)* | 1.24 (1.05 to 1.46)* |
| 15 years | 1.71 (1.38 to 2.13)*** | 1.41 (1.16 to 1.71)** |
| ≥16 years | 2.18 (1.76 to 2.72)*** | 1.34 (1.07 to 1.69)* |
| Rural (ref:urban) | 0.97 (0.73 to 1.27) | 0.99 (0.85 to 1.15) |
| Types of school (ref:middle school) | | |
| Academic high school | 0.79 (0.67 to 0.93)* | 0.61 (0.49 to0.77)*** |
| Vocational high school | 5.09 (4.05 to 6.40)*** | 3.71 (2.83 to 4.85)*** |
| Parental marital status (ref:married) | | |
| Others | 1.52 (1.37 to 1.69)*** | 1.26 (1.13 to 1.41) *** |
| Paternal education (ref: middle or below) | | |
| High school | 0.75 (0.68 to 0.82)*** | 0.88 (0.81 to 0.98)* |
| College or above | 0.40 (0.33 to 0.48)*** | 0.69 (0.58 to 0.82)*** |
| Unknown | 0.98 (0.85 to 1.13) | 0.83 (0.68 to 1.01) |
| Maternal education (ref: middle or below) | | |
| High school | 0.75 (0.67 to 0.83)*** | 0.90 (0.82 to 1.00) |
| College or above | 0.38 (0.33 to 0.45)*** | 0.66 (0.58 to 0.76)*** |
| Unknown | 1.14 (1.00 to 1.30)* | 1.18 (0.99 to 1.41) |
| Academic performance (ref:excellent) | | |
| Middle | 1.55 (1.42 to 1.69)*** | 1.36 (1.25 to 1.48)*** |
| Poor | 2.35 (2.12 to 2.60)*** | 2.07 (1.86 to 2.30)*** |
| Self-reported health (ref:very good/good) | | |
| Fair | 1.39 (1.30 to 1.49)*** | 1.14 (1.07 to 1.22)*** |
| Bad/very bad | 1.59 (1.38 to 1.84)*** | 1.31 (1.14 to 1.49)*** |
| Unknown | 1.46 (1.07 to 2.00)* | 1.11 (0.78 to 1.57) |
| Loneliness (ref:never/occasionally) | | |
| Sometimes | 1.30 (1.20 to 1.40)*** | 1.20 (1.09 to 1.32)*** |
| Often/always | 1.38 (1.24 to 1.54)*** | 1.20 (1.08 to 1.34)** |
| Physical activity (ref: 0 day/week) | | |
| 1–2 days/week | 0.81 (0.74 to 0.89)*** | 0.90 (0.81 to 0.99)* |
| 3–5 days/week | 0.74 (0.68 to 0.82)*** | 0.91 (0.83 to 0.99)* |
| 6–7 days/week | 0.69 (0.61 to 0.79)*** | 0.93 (0.82 to 1.05) |
| Breakfast (ref:0 day/week) | | |
| 1–2 days/week | 1.38 (1.03 to 1.85)* | 1.25 (0.92 to 1.70) |
| 3–4 days/week | 1.21 (0.97 to 1.51) | 1.04 (0.81 to 1.33) |
| ≥5 days/week | 0.70 (0.57 to 0.86)** | 0.95 (0.74 to 1.22) |
| Fruit (ref:1–2 times/day) | | |
| <1 times/day | 2.36 (2.09 to 2.66)*** | 1.44 (1.29 to 1.60)*** |
| ≥3 times/day | 2.01 (1.65 to 2.44)*** | 1.18 (1.01 to 1.38)* |
| Vegetable (ref:1–2 times/day) | | |
| <1 time/day | 1.28 (1.13 to 1.45)*** | 0.99 (0.86 to 1.14) |
| ≥3 times/day | 1.27 (1.16 to 1.39)*** | 0.98 (0.90 to 1.07) |
| Carbonated beverages (ref:none) | | |
| 1–6 times/week | 1.02 (0.95 to 1.09) | 1.02 (0.96 to 1.09) |

Continued

**Table 4** Continued

| Characteristics | COR (95% CI) | AOR (95% CI)† |
|---|---|---|
| 1–2 times/day | 0.98 (0.83 to 1.15) | 0.99 (0.84 to 1.18) |
| ≥3 times/day | 1.50 (1.25 to 1.80)*** | 1.29 (1.03 to 1.60)* |

*P<0.05, **P<0.001, ***P<0.0001.
†Adjusted for all other covariates listed in the table.
AOR, adjusted OR; COR, crude OR.

at middle and academic high schools. Hence, they might have more time available to spend on electronic screen products.

### Relationship of high ST with its correlates

The inverse association between parental education level and high ST observed in our study may be due to the fact that highly educated parents were more inclined to limit children's ST at home than less highly educated parents. In China, it is estimated that approximately 50% of families have no specific rules for ST,[7] but having ST rules at homes might have a protective effect on children's excessive ST.[7 32] In addition, students living in non-intact families and those often or always feeling lonely had much higher odds of high ST, which is consistent with previous studies.[33] These findings suggest parental care and company are important for this age group in terms of controlling high ST and improving academic performance, because academic performance was inversely associated with high ST, consistent with the findings of previous studies.[15 34] We found that poor self-reported health was positively associated with high ST. This might reflect bad health preventing students from engaging in physical activity, with an associated increased likelihood of excessive electronic screen product exposure. Another possible reason might be that excessive electronic screen product exposure could have a negative impact on the health of adolescents.

As expected, being physically active was negatively associated with high ST in our study. A previous study found that junior school students in China lacked awareness of the importance of physical activity,[35] despite government guidelines suggesting adolescents should undertake at least 60 min of physical activity daily.[36] It is possible that reducing ST would eventually increase physical activity levels with associated psychological benefits and improved quality of life, academic performance and self-esteem.[37 38] Our study demonstrated that only about 16.7% students reported being physically active every day during the past 7 days, which was lower than the average level in China (22.7%),[39] suggesting action is needed to increase physical activity levels among adolescents in Zhejiang.

Dietary guidelines in China recommend consumption of 200–350 g of fruit and 300–500 g of vegetables daily. It is not possible to estimate exact quantities of fruit and vegetables consumed daily through a self-administrated survey among adolescents, but our study found over 70% of students consumed fruits once daily or more

frequently, higher than the proportion (64%) among adolescents in the USA.[40] A previous study found TV viewing was inversely related to intake of fruit (OR 0.92) and vegetables (OR 0.95) among the US adolescents.[41] In our study, compared with students consuming fruits 1–2 times per day, those consuming fruits less than once a day had a higher likelihood of high ST. Interestingly, those consuming fruits ≥3 times per day also had a higher probability of high ST, in contrast with results from a previous study in the USA[41] in which fruit intake was divided into two groups ('≥1 time per day' and '<1 time per day'). One possible explanation for this U-shaped association might be that some adolescents with excessive exposure to electronic screen may consume excessive quantities of fruit. In the present study, half of the students consumed vegetable ≥3 times every day, higher than the USA (18.5%).[42] Although vegetable intake was negatively associated with high ST in univariate logistic regression, there was no statistically significant association in multivariable analyses.

Many studies have demonstrated associations of sugar-sweetened beverages (SSB) with chronic diseases (eg, obesity, diabetes, hypertension, stroke).[43–46] Schulze et al found that individuals consuming ≥1 SSB per day had an 83% higher risk of developing type 2 diabetes compared with those consuming <1 SSB per month.[43] In our study, 6.9% of students reported consuming carbonated beverages ≥1 time per day. Although the percentage was far lower than the USA (20.4%),[47] this raises concerns about SSB consumption among adolescents in China. Gebremariam et al found that an increase in TV viewing by an hour was associated with the consumption of 30 mL more soft drinks in Greece and 90 mL more soft drinks in Switzerland.[16] The present study showed that students consuming carbonated drinks ≥3 times per day have about 30% higher risk of high ST, which was consistent with a previous study.[48] It is possible that students with high ST might be more frequently exposed to food advertisements, which are often for unhealthy foods, including carbonated beverages.[49]

### Implications

With the development of internet technology and the emergence of new screen products, it is inevitable that adolescents will have more opportunity to spend time using electronic screen products. Our study has several important implications. First, excessive electronic screen product exposure appears to be an increasingly common

behaviour among students in Zhejiang, and without further intervention, it will become more common over coming decades. Second, comprehensive intervention measures, including strict rules on duration of using electronic products at home and increasing physical activity need to be taken into account; these interventions might also benefit the development of healthy dietary habits, improving physiological and psychological health.

## Strengths and limitations

The strengths of this study include the large provincially representative sample, high response rate and use of standardised procedures. In addition, the study questionnaire included a large number of sociodemographic and behavioural risk factors. There are also, however, several limitations. First, the cross-sectional study design prevents establishment of the causal relationship of these factors with high ST. Second, the questionnaire focused on aggregated ST, and did not allow investigation of time spent on specific screen products (eg, TV, computer, video game and mobile phone). Third, all data were self-reported by students and not objectively measured, which might increase the risk of information bias. Fourth, we only collected information on ST on school days, and did not include non-school days. Given that students usually spent more time on screen products on non-school days,[7] the prevalence of high ST observed in our study might be an underestimate.

## CONCLUSIONS

In summary, our study extended existing literature by describing the patterns and associations of high ST among a provincial representative sample of adolescents in China, and found that high ST is prevalent among students and associated with a cluster of sociodemographic and unhealthy behavioural risk factors in Zhejiang, China.

**Acknowledgements** The authors would like to thank all the students, parents, teachers and local officials for their participation, assistance and cooperation.

**Contributors** HW and MY designed the study. HW collected and analysed data and wrote manuscript. JZ and RH involved in data interpretation. HD and BF provided critical comments on the manuscript. All the authors read and approved the final submitted version.

**Funding** The work was supported by grant (2016YFC0900502) from National Key Research and Development Program of China.

**Competing interests** None declared.

**Patient consent** Not required.

**Ethics approval** The study design and procedure was approved by the ethics committee of Zhejiang Provincial CDC.

**Provenance and peer review** Not commissioned; externally peer reviewed.

**Data sharing statement** No additional data are available.

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
