## [Reviewer comments · BMJ Open]

ARTICLE DETAILS

TITLE (PROVISIONAL)	Prevalence of high screen-time and associated factors among students: a cross-sectional study in Zhejiang, China
AUTHORS	Wang, Hao; Zhong, Jie-Ming; Hu, Ru-Ying; Bragg, Fiona; Yu, Min; Du, Huaidong

VERSION 1 – REVIEW

REVIEWER	Hiroyuki Hikichi Harvard T.H.CHAN School of Public Health, USA
REVIEW RETURNED	24-Jan-2018

GENERAL COMMENTS	The authors explored epidemiological characteristics that explained the prevalence of screen time among Chinese adolescents. They showed clear associations. 1) Can you separate the outcome into each product and run the same analysis (i.e., 1: TV; 2: play pad; 3: games; 4: phone; and 5: personal computer)? For example, a systematic review showed that mobile devices such as smartphones cause sleep problems of adolescents. Sleep Med Rev. 2015 Jun;21:50-8. doi: 10.1016/j.smrv.2014.07.007. Epub 2014 Aug 12. Screen time and sleep among school-aged children and adolescents: a systematic literature review. Hale L, Guan S. From the study, each product may have unique influences on their health problems. And, I suppose that there is a gap in possession rate of each product between rich households and poor households. From these reasons, I want to see the result of each product. 2) Can you control problems of relationships with parents or friends in the multivariate regression?
--

REVIEWER	Daniela Husarova, PhD Department of Health Psychology, Faculty of Medicine, P.J. Safarik University in Kosice, Slovak Republic
REVIEW RETURNED	02-Feb-2018

GENERAL COMMENTS	This article investigate current problem among adolescent which became more and more common over the world. The aim of this article is processed appropriate and adds more information in this field of research among adolescent in Chinese population. However, I have some issues which need to be solved.
---

Introduction

The scientific background of article is explained quite well, but I miss more information about consequences of excessive screen time. Authors should spend more words on associations of excessive screen time and health (e.g. psychological and physiological health complaints, such as headache, backache, sleeping difficulties,...). Moreover, authors should mention also other factors which might be associated with excessive screen-based behavior, such as family environment (parental rules, joint family activities,..), or peers. For that purpose I recommend articles:
Brindova D., Dankulincova Veselska Z., Klein D., Hamrik Z., Sigmundova D., van Dijk J.P., Reijneveld S.A., Madarasova Geckova A.: Is the association between screen-based behaviour and health complaints among adolescents moderated by physical activity? *International Journal of Public Health*, 2015; 60(2):139-145.

Brindova D., Pavelka J., Sevcikova A., Zezula I., van Dijk J.P., Reijneveld S.A., Madarasova Geckova A.:How parents can affect excessive spending of time on screen-based activities. *BMC Public Health*, 2014, 14:1261.

Husarova D, Blinka L, Madarasova Geckova A, Sirucek J, van Dijk JP, Reijneveld SA. Do sleeping habits mediate the association between time spent on digital devices and school problems in adolescence? *European Journal of Public Health*, 2017.

P4, L8 – I would skip this sentence as I missed the connection to previous sentences and it is also mentioned in next paragraph connected to association of ST and chronic conditions.

Methods

I would prefer rename “Survey design” on “Sample and procedure” and in this case, to join the text from Quality control as well as Ethics statement to description of study procedure. Particularly, after the sentence “All the students in the chosen classes were invited to participate in the survey.” (P5, L4-5) should be insert the sentence “Written informed consent was given before survey, and obtained from all participants and their guardians”. (P5, L41-42). Sentence about ethics committee add to the end of the Sample and procedure. Moreover, text from “Quality control” move after the sentence “Subjects filled in the anonymous self-administrated questionnaire in the classrooms” (P5, L12-13). In addition, I would avoid term subjects, respondents is more appropriate.

Results

Description of the sample, P6, L16-24, should be placed in “Sample and procedure” after the sentence “In stage two, 10 classes of middle school, 5 classes of academic high school, and 5 classes of vocational high school were selected randomly within each chosen counties, respectively.”

Tables

I have only minor comments. please avoid using abbreviation in titles of tables.

Table 1 – answer option should be in separate column for better clarity

Table 2 – just add N(%) below “Total, Boys, Girls” to show that authors are presenting particular number and percentage of prevalence.

Table 4 – for confidence intervals use *p<0.05, **p<0.001, ***p<0.001.

	Discussion Start the discussion with authors main findings and then discuss them with previous findings. Authors should mentioned also strengths of this study and possible implications for future research and practice.
--	---

VERSION 1 – AUTHOR RESPONSE

Responses to the Journal requirements

1. Please work to improve the quality of the English throughout your manuscript. We strongly recommend asking a native English speaking colleague to assist you or enlisting the help of a professional copy-editing service.

Response: Done. The article was polished by a native English speaker named Bragg Fiona.

2. Please revise your title to indicate the research question, study design, and setting. This is the preferred format of the journal.

Response: Done. The title has been revised as follows “Prevalence of high screen-time and associated factors among students: a cross-sectional study in Zhejiang, China ”.

3. In the methods section please elaborate on the measurement of screen time used in this study. For example, has the measure been assessed for reliability and validity?

Response: Thanks for this suggestion. Detailed information on outcome variable and covariates were added in the method section and Table 1. The validity and reliability of questionnaire were not conducted among students from Zhejiang, but US CDC ever conducted one test-retest reliability studies of the national YRBS questionnaire in 1992. the 1991 version of the questionnaire was administered to a convenience sample of 1,679 students in grades 7–12. The questionnaire was administered on two occasions, 14 days apart. Approximately three fourths of the questions were rated as having a substantial or higher reliability ($\kappa = 61\%–100\%$), and no statistically significant differences were observed between the prevalence estimates for the first and second times that the questionnaire was administered. No study has been conducted to assess the validity of all self-reported behaviors that are included on the YRBSS questionnaire. (Centers for Disease Control and Prevention. Methodology of the Youth Risk Behavior Surveillance System-2013. MMWR Recomm Rep. 2013, 62 (RR-1):1-20.). It is a good idea! We are considering to test validity and reliability of adolescent risk behavior questionnaire among students from Zhejiang.

Responses to Reviewer #1

1. Can you separate the outcome into each product and run the same analysis (i.e., 1: TV; 2: play pad; 3: games; 4: phone; and 5: personal computer)? For example, a systematic review showed that mobile devices such as smartphones cause sleep problems of adolescents.

Sleep Med Rev. 2015 Jun;21:50-8. doi: 10.1016/j.smrv.2014.07.007. Epub 2014 Aug 12.

Screen time and sleep among school-aged children and adolescents: a systematic literature review. Hale L, Guan S.

From the study, each product may have unique influences on their health problems. And, I suppose that there is a gap in possession rate of each product between rich households and poor households. From these reasons, I want to see the result of each product.

Response: Thanks for good suggestion. I absolutely agree with reviewer’s opinion that each product may have unique influences on adolescent health. However, the time spent on each electronic product was not included in the design process of questionnaire. Your suggestion is very instructive and point out the future research direction. We added this point in the limitation section.

3. Can you control problems of relationships with parents or friends in the multivariate regression?

Response: Thanks for good suggestion. Although our questionnaire was designed based on the well-known YRBS conducted by US CDC and the international GSHS supported by WHO, in which relationships with parents or friends was not included, In the next version, we will add related questions in the our own questionnaire. Thanks!

Responses to Reviewer #2

1. Introduction

The scientific background of article is explained quite well, but I miss more information about consequences of excessive screen time. Authors should spend more words on associations of excessive screen time and health (e.g. psychological and physiological health complaints, such as headache, backache, sleeping difficulties,...). Moreover, authors should mention also other factors which might be associated with excessive screen-based behavior, such as family environment (parental rules, joint family activities,..), or peers. For that purpose I recommend articles:

Brindova D., Dankulinova Veselska Z., Klein D., Hamrik Z., Sigmundova D., van Dijk J.P., Reijneveld S.A., Madarasova Geckova A.: Is the association between screen-based behaviour and health complaints among adolescents moderated by physical activity? *International Journal of Public Health*, 2015; 60(2):139-145.

Brindova D., Pavelka J., Sevcikova A., Zezula I., van Dijk J.P., Reijneveld S.A., Madarasova Geckova A.: How parents can affect excessive spending of time on screen-based activities. *BMC Public Health*, 2014, 14:1261.

Husarova D, Blinka L, Madarasova Geckova A, Sirucek J, van Dijk JP, Reijneveld SA. Do sleeping habits mediate the association between time spent on digital devices and school problems in adolescence? *European Journal of Public Health*, 2017.

P4, L8 – I would skip this sentence as I missed the connection to previous sentences and it is also mentioned in next paragraph connected to association of ST and chronic conditions.

Response: Three articles are very instructive, and were added in introduction section and discussion section.

2. Methods

I would prefer rename "Survey design" on "Sample and procedure" and in this case, to join the text from Quality control as well as Ethics statement to description of study procedure. Particularly, after the sentence "All the students in the chosen classes were invited to participate in the survey." (P5, L4-5) should be insert the sentence "Written informed consent was given before survey, and obtained from all participants and their guardians". (P5, L41-42). Sentence about ethics committee add to the end of the Sample and procedure.

Moreover, text from "Quality control" move after the sentence "Subjects filled in the anonymous self-administrated questionnaire in the classrooms" (P5, L12-13). In addition, I would avoid term subjects, respondents is more appropriate.

Response: Done

3. Results

Description of the sample, P6, L16-24, should be placed in "Sample and procedure" after the sentence "In stage two, 10 classes of middle school, 5 classes of academic high school, and 5 classes of vocational high school were selected randomly within each chosen counties, respectively."

Response: Done

4. Tables

I have only minor comments. please avoid using abbreviation in titles of tables.

Response: Done

Table 1 – answer option should be in separate column for better clarity

Response: Done

Table 2 – just add N(%) below "Total, Boys, Girls" to show that authors are presenting particular number and percentage of prevalence.

Response: Done

Table 4 – for confidence intervals use * $p < 0.05$, ** $p < 0.001$, *** $p < 0.001$.

Response: Done

5. Discussion

Start the discussion with authors main findings and then discuss them with previous findings.

Authors should mentioned also strengths of this study and possible implications for future research and practice.

Response: Done

6.FORMATting AMENDMENTS (if any)

Required amendments will be listed here; please include these changes in your revised version:

1.Article Summary

- Please ensure that Article Summary is placed after the abstract.

Response: Summary was added and is placed after the abstract.

VERSION 2 – REVIEW

REVIEWER	Hiroyuki Hikichi Harvard T. H. CHAN School of Public Health
REVIEW RETURNED	07-Apr-2018

GENERAL COMMENTS	There are no additional comments.
-----------------------------------

REVIEWER	Dr. Daniela Husarova Department of Health psychology Faculty of Medicine PJ Safarik University Kosice Slovakia
REVIEW RETURNED	20-Apr-2018

GENERAL COMMENTS	Thank you for this article and I hope that you will continue in this interesting topic.
---